# Context Window Failures in Relational Foundation Models

**Denis Oliveira Correa** [1]  **Francisco Galuppo Azevedo** [1 2]

## Abstract

Recent Relational Deep Learning architectures have been proposed as foundation models for multi-table relational data, yet they impose constrained neighborhood budgets that force row truncation when an entity has many related records. We introduce Animus, a synthetic financial dataset in which predicting customer income requires aggregating up to tens of thousands of transactions. On the raw representation, three recently proposed models (RT, Griffin, RelGT) achieve $R^2 \leq 0.18$; a single, routine, temporal pre-aggregation step recovers $R^2$ up to $0.65$. This questions whether current relational foundation models are ready for high-cardinality real-world data.

## 1. Introduction

Relational databases are the dominant data format in industry, underpinning applications from fraud detection to credit scoring. Predictive tasks on such data require joining and aggregating across multiple entity and event tables, a setting that has resisted the unified modeling that foundation models have achieved in language and vision. Robinson et al. (2024) formalized this challenge as a benchmark, spurring a wave of architectures that claim foundation-model status for relational data.

All three recently proposed foundation model architectures impose a hard limit on how many rows per entity can be incorporated. Relational Transformer (RT) (Ranjan et al., 2026) is bounded by the context size; Relational Graph Transformer (RelGT) (Dwivedi et al., 2026) is bounded by its subgraph sampling budget; and Griffin (Wang et al., 2025) is limited by its per-hop fanout. In all cases, rows/cells beyond the budget are discarded. In real-world relational data, however, high-frequency entities can accumulate or-

ders of magnitude more records than any of these budgets allow. This gap has not been stress-tested.

We introduce Animus, a synthetic financial dataset designed to stress exactly this limit. Customer transaction frequency is heterogeneous: $60\%$ of customers are salaried and generate roughly one transaction per month, while $5\%$ are high-volume and generate between one thousand and ten thousand. We evaluate RT, Griffin, RelGT, and GraphSAGE (Hamilton et al., 2017) on two representations of the same data, raw individual events and a simple monthly aggregation, and find that the three budget-limited models recover up to $+0.47$ $R^2$ when given pre-aggregated input, while GraphSAGE, which employs a much larger sampling budget, already performs well on raw data and improves by only $+0.06$.

The rest of this paper is organized as follows. Section 2 reviews the evaluated models. Section 3 describes Animus and the prediction task. Section 4 presents results and analysis. Section 5 discusses implications. Dataset generation code and model configurations will be made publicly available upon acceptance.

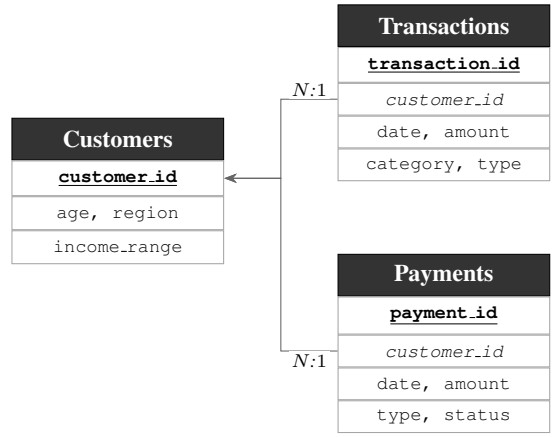

*Figure 1.* Relational schema of Animus.

## 2. Background

The Relational Deep Learning paradigm, formalized by Fey et al. (2024), represents a relational database as a heterogeneous temporal graph: each row is a node, and foreign-key relationships are directed edges. Tasks are defined at the en-

[1]Kunumi Institute, Brazil [2]Universidade Federal de Minas Gerais, Belo Horizonte, Minas Gerais, Brazil. Correspondence to: Francisco Galuppo Azevedo <franciscogaluppo@dcc.ufmg.br>.

*2nd ICML Workshop on Foundation Models for Structured Data (FMSD @ ICML 2026)*, Seoul, South Korea, 2026. Copyright 2026 by the author(s).

tity level and evaluated at a fixed timestamp, so each model must aggregate information from an entity's neighborhood within the graph. We evaluate four models:

- **GraphSAGE** (Hamilton et al., 2017) is a neighborhood-aggregation GNN that applies a permutation-invariant aggregator (sum) over sampled neighbors.
- **RT** (Ranjan et al., 2026) is a foundation model pre-trained with masked token prediction that tokenizes data cell-wise and applies attention along three axes: row-wise (across records), column-wise (across features), and across primary-foreign key links; the number of cells is bounded by `seq_len`.
- **Griffin** (Wang et al., 2025) is a graph-centric GNN that propagates information via message passing with a small per-hop fanout.
- **RelGT** (Dwivedi et al., 2026) uses graph structure to sample a subgraph, enriches each sampled node with metadata (type, hop distance, time, local structure), and processes the resulting tokens through a transformer with local and global attention.

All three budget-limited models discard records beyond their limit regardless of how informative those records might be. GraphSAGE operates under a larger but still finite sampling budget and would degrade at sufficiently extreme cardinalities. The question we ask is whether the budgets imposed by current architectures are insufficient for realistic relational data.

## 3. Method

Animus is a synthetic financial dataset comprising $M = 100\,000$ customers. Customer registrations are spread over January 2023 to January 2024; each customer accumulates twelve months of transaction history from their registration date. The database has three relational tables linked by `customer_id`: **Customers**, **Transactions**, and **Payments** (Figure 1). Because a small fraction of customers generate thousands of transactions per month, the total transaction volume reaches tens of millions of rows; node-count statistics are reported in Appendix D.

**Prediction task**

The task is entity-level income regression defined within the RelBench framework (Robinson et al., 2024). Given all data up to a prediction timestamp $t$, the model must estimate each customer's total credit income in the month following $t$. We use three temporal splits: training cutoff 2024-03-31, validation cutoff 2024-06-30, and test cutoff 2024-09-30. No transactions after $t$ are visible at prediction time.

**Customer income**

Each customer $i$ is assigned a base monthly income $y_i^{\text{base}} \in \{\$1,000, \$2,000, \$3,000\}$ with probabilities $(0.50, 0.35, 0.15)$. Monthly income is not constant, total income over the customer's twelve-month tenure is distributed across months according to one of four temporal income patterns, assigned at registration:

- Uniform (50%): equal weight across all twelve months;
- Back-loaded (20%): weights $(0.1, 0.2, 0.3, 0.4)$ on the final four months, zero elsewhere;
- Front-loaded (15%): weights $(0.4, 0.3, 0.2, 0.1)$ on the first four months, zero elsewhere;
- Mid-peak (15%): weights $0.5$ and $0.3$ on the two middle months, $0.05$ on all others (then normalized).

Let $y_i^{(m)}$ denote customer $i$'s ground-truth income in month $m$; the prediction target is then $y_i^{(m+1)}$.

**Credit transactions**

Each customer is assigned a transaction frequency class at registration (Figure 2):

- Single (60%): exactly 1 credit per month;
- Low (25%): $K_i^{(m)} \sim U[4, 10]$ credits per month;
- Medium (10%): $K_i^{(m)} \sim U[100, 1,000]$;
- High ( 5%): $K_i^{(m)} \sim U[1,000, 10,000]$.

Within month $m$, $K_i^{(m)}$ individual credit amounts are generated by drawing a weight vector $\mathbf{w} \sim \text{Dirichlet}(\mathbf{1}_K)$ and setting each amount to $w_k \cdot y_i^{(m)}$, so the credits in that month sum *exactly* to $y_i^{(m)}$. Transaction categories are class-dependent: *single* draws from {`salary`, `freelance`} with probabilities $(0.8, 0.2)$; *low* from {`salary`, `freelance`, `bonus`} with $(0.5, 0.4, 0.1)$; *medium* and *high* draw uniformly from all five categories {`salary`, `freelance`, `bonus`, `refund`,

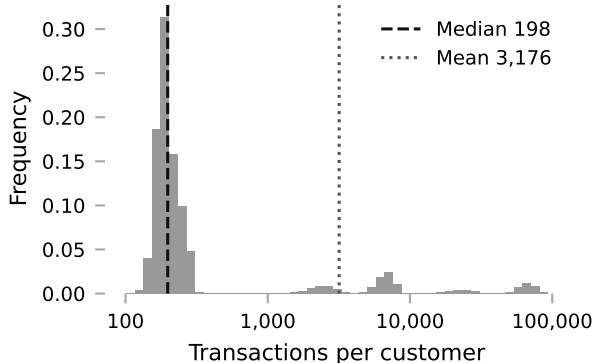

*Figure 2.* Transaction count distribution per customer (log scale).

transfer_in}. Since the target $y_i^{(m+1)}$ equals the sum of all $K_i^{(m+1)}$ credit amounts, a model that observes only $k < K_i^{(m+1)}$ of them obtains an expected estimate of $y_i^{(m+1)} \cdot k/K_i^{(m+1)}$, an underestimate that grows worse as $K_i^{(m+1)}$ increases. This is the core adversarial property of the benchmark.

### Debit transactions

Each customer receives $D_i^{(m)} \sim U[5,\, 24]$ debit transactions per month, independent of frequency class, clipped downward when the monthly debit budget would make that many transactions infeasible. A per-customer balance ratio $\rho_i$ is drawn once at registration from one of three regimes:

- Balanced: $\rho_i \sim U[0.95,\, 1.05]$ (35%);
- Deficit: $\rho_i \sim U[1.10,\, 1.50]$ (15%);
- Surplus: $\rho_i \sim U[0.40,\, 0.90]$ (50%).

Total debits over the tenure equal $\rho_i \cdot \sum_m y_i^{(m)}$, distributed across months by drawing $U[0.7,\, 1.3]$ multipliers and normalizing. Individual debit amounts are sampled sequentially subject to the monthly budget, with bracket probabilities: small \$5–\$100 (60%), medium \$100–\$500 (30%), large \$500–\$5,000 (10%). Spending categories are assigned by realized amount:

- More than \$1,000: {rent 60%, shopping 30%, healthcare 10%};
- \$200–\$1,000: {shopping 40%, utilities 30%, entertainment 20%, transportation 10%};
- Less than \$200: {groceries 40%, dining 30%, transportation 20%, shopping 10%}.

### Payments

A uniformly random 30–40% of customers have payment records representing debt-service obligations (loans, mortgages, credit cards). Each such customer has $U[50,\, 599]$ payment events with amounts drawn from $\mathrm{LogNormal}(\mu=\log 1{,}700,\, \sigma=0.8)$ clipped to [\$50, \$5,000], type drawn uniformly from {loan, credit_card, mortgage, auto_loan}, and dates drawn uniformly over the customer's tenure. Payments are not part of the income label and carry no predictive signal for it: payment customers are selected uniformly at random without conditioning on income, and payment amounts are drawn from a fixed distribution independent of income_range. They serve solely to add relational realism to the schema.

### Two representations

We evaluate every model on two representations of the same underlying data.

**Raw.** Each credit and debit event is a separate transaction node; the largest customers have up to $88\,717$ transaction nodes (Appendix D).

**Agg-Simple.** Transactions are collapsed into monthly credit and debit sums per spending category (table txn_monthly_cat), and payments into monthly totals (table pay_monthly). The maximum neighborhood size becomes $12 \times C$ rows, where $C$ is the number of distinct categories.

Both representations encode the same aggregate information; the only difference is cardinality. The same hyperparameter grid is applied to both representations with no model-specific tuning.

## 4. Experiments

We evaluate all four models on both representations using a hyperparameter grid search over learning rate and architecture depth, following the same search spaces used in each model's original paper. All experiments run on a single NVIDIA H200 GPU. For RT we search num_blocks $\in \{6, 12\}$ and lr $\in \{3 \times 10^{-5}, 10^{-4}, 3 \times 10^{-4}\}$; for Griffin we search num_mp $\in \{2, 4, 6\}$ and lr $\in \{10^{-4}, 3 \times 10^{-4}, 10^{-3}\}$; for RelGT we search num_layers $\in \{4, 8\}$ and lr $\in \{10^{-4}, 3 \times 10^{-4}, 8 \times 10^{-4}\}$. RT runs with seq_len=1024. The primary evaluation metric is $R^2$; MAE and RMSE are reported as secondary metrics. Full grid search tables are in Appendix B.

Results are shown in Table 1. On the raw representation, RT, Griffin, and RelGT score $R^2 \leq 0.18$, with RelGT failing to construct the graph entirely due to memory constraints. GraphSAGE reaches $R^2 = 0.69$. Switching to agg-simple, RT improves from $0.18$ to $0.65$ ($+0.47$) and Griffin from $0.11$ to $0.38$ ($+0.27$), confirming the neighborhood budget hypothesis. GraphSAGE improves only modestly ($0.69 \rightarrow 0.75$), consistent with it already aggregating a large sample of neighbors on raw data. Full metrics including MAE and RMSE are in Appendix A.

*Table 1.* Test $R^2$ (higher is better) on income prediction. Best result per model from hyperparameter grid search. OOM: graph materialisation ran out of memory.

| Model | Raw | Agg-Simple |
|---|---|---|
| GraphSAGE | 0.693 | **0.752** |
| RT | 0.184 | 0.654 |
| Griffin | 0.110 | 0.384 |
| RelGT | OOM | 0.230 |

Table 2 localizes the failure by stratifying GraphSAGE performance on the $N = 474$ customers whose post-cutoff credit activity diverges from their training history, which is why all per-bin $R^2$ values fall below the overall 0.69 to

0.75 range. For low-frequency customers (bins 1–2, $\leq 16$ extra transactions) raw is marginally better, as fine-grained event patterns carry signal that aggregation discards. For high-frequency customers (bins 3–4, $\geq 16$ extra transactions) agg-simple wins by $+0.44$ $R^2$ in both bins. The crossover confirms that the failure is localized to customers whose neighborhood exceeds the models' sampling budgets. Extended per-bin metrics are in Appendix C.

*Table 2.* GraphSAGE test $R^2$ stratified by post-cutoff credit transaction count ($N = 474$ of 10,000 test customers with changed post-cutoff behavior; overall Raw = 0.692, Agg-Simple = 0.752). Per-bin values are lower than the overall because the bins isolate the hardest cases.

| Txn diff | N | $R^2$ **Raw** | $R^2$ **Agg** | $\Delta R^2$ |
|---|---|---|---|---|
| 1–5 | 280 | 0.636 | 0.529 | $-0.107$ |
| 5–16 | 76 | 0.500 | 0.340 | $-0.160$ |
| 16–688 | 44 | 0.165 | 0.608 | $+0.443$ |
| 688–15,336 | 74 | 0.063 | 0.506 | $+0.443$ |

The *agg-simple* representation is a SQL aggregation that any analyst would apply as routine data preparation. Recovering 0.2–0.5 $R^2$ from it is a brittle outcome for something marketed as a foundation model. A foundation model should learn what to aggregate, not rely on the practitioner to do it first.

## 5. Conclusion

We introduced Animus, a synthetic financial benchmark designed to expose neighborhood budget limitations in Relational Deep Learning models. On the raw representation, three recently proposed models achieve $R^2 \leq 0.18$, while a routine temporal aggregation step recovers up to $R^2 = 0.65$. Our per-bin analysis traces the failure to customers whose post-cutoff transaction count exceeds the models' sampling budgets.

Animus can be extended with a *needle-in-a-haystack* variant that would replace the bulk-aggregation task with targeted retrieval. The label would depend on a single specific transaction (e.g., a fraudulent charge) embedded among thousands of normal ones, testing whether models can perform selective attention over long relational context.

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

# A. Full Metrics for Best Configurations

*Table 3.* Test MAE, RMSE, and $R^2$ for the best hyperparameter configuration of each model on each dataset representation. RT did not log MAE/RMSE; RMSE not available for Griffin and RelGT grid-search configurations. OOM: graph materialisation out of memory.

| Model | Dataset | MAE | RMSE | $R^2$ |
|---|---|---|---|---|
| GraphSAGE | Raw | 369.63 | 604.71 | 0.6925 |
| GraphSAGE | Agg-Simple | 278.55 | 543.59 | 0.7515 |
| RT | Raw | — | — | 0.1836 |
| RT | Agg-Simple | — | — | 0.6541 |
| Griffin | Raw | 711.39 | — | 0.1101 |
| Griffin | Agg-Simple | 595.12 | — | 0.3836 |
| RelGT | Raw | | OOM | |
| RelGT | Agg-Simple | 606.92 | — | 0.2299 |

# B. Grid Search Results

### RelGT

Fixed config: `channels=512`, `num_heads=4`, `ff_dropout=0.3`, `attn_dropout=0.3`, `num_neighbors=300`, `batch_size=256`, `epochs=100`. Raw dataset was not evaluated due to OOM during graph materialisation.

*Table 4.* RelGT grid search on the **agg-simple** dataset. Bold = best test $R^2$.

| lr | num_layers | Best val $R^2$ | Test $R^2$ | Test MAE |
|---|---|---|---|---|
| **1e-4** | **4** | **0.277** | **0.230** | **606.92** |
| 3e-4 | 4 | 0.306 | 0.206 | 626.25 |
| 1e-4 | 8 | 0.324 | 0.192 | 618.31 |
| 8e-4 | 8 | 0.303 | 0.190 | 620.95 |
| 3e-4 | 8 | 0.229 | 0.179 | 624.65 |
| 8e-4 | 4 | 0.275 | 0.136 | 653.50 |

### Relational Transformer (RT)

Fixed config: `d_model=256`, `seq_len=1024`, `max_steps=32769`, `batch_size=32`, `wd=0.0`.

*Table 5.* RT grid search on the **raw** dataset. Bold = best test $R^2$.

| lr | num_blocks | Best val $R^2$ | Test $R^2$ |
|---|---|---|---|
| 3e-5 | 12 | 0.077 | 0.039 |
| 1e-4 | 12 | 0.220 | −0.552 |
| **3e-4** | **12** | **0.584** | **0.184** |
| 3e-5 | 6 | 0.226 | 0.118 |
| 1e-4 | 6 | 0.323 | −0.126 |
| 3e-4 | 6 | 0.393 | 0.086 |

*Table 6.* RT grid search on the **agg-simple** dataset. Bold = best test $R^2$.

| lr | num_blocks | Best val $R^2$ | Test $R^2$ |
|---|---|---|---|
| 3e-5 | 12 | 0.045 | 0.042 |
| 1e-4 | 12 | 0.620 | 0.025 |
| 3e-4 | 12 | 0.694 | 0.595 |
| 3e-5 | 6 | 0.084 | $-0.084$ |
| 1e-4 | 6 | 0.473 | $-0.251$ |
| **3e-4** | **6** | **0.720** | **0.654** |

## Griffin

Fixed config: `hiddim=512, hop=2, fanout=20, wd=0.0002, epochs=50, batchsize=256`.

*Table 7.* Griffin grid search on the **raw** dataset (partial). The 31M-transaction neighbourhood-mapping step made some runs exceed the per-run time limit; results below are from completed runs. Bold = best test $R^2$.

| lr | num_mp | Best val $R^2$ | Test $R^2$ | Test MAE |
|---|---|---|---|---|
| **3e-4** | **6** | **0.204** | **0.110** | **711.39** |
| 1e-4 | 4 | 0.174 | 0.071 | 749.82 |
| 1e-3 | 4 | 0.177 | 0.066 | 756.48 |
| 1e-3 | 2 | 0.195 | 0.062 | 742.68 |
| 3e-4 | 4 | 0.178 | 0.051 | 767.90 |
| 1e-3 | 6 | 0.158 | 0.015 | 808.87 |
| 1e-4 | 2 | 0.088 | 0.012 | 789.91 |
| 3e-4 | 2 | 0.152 | $\approx 0.000$ | 817.89 |
| 1e-4 | 6 | 0.156 | $-0.010$ | 804.11 |

*Table 8.* Griffin grid search on the **agg-simple** dataset. Bold = best test $R^2$.

| lr | num_mp | Best val $R^2$ | Test $R^2$ | Test MAE |
|---|---|---|---|---|
| **1e-3** | **6** | **0.552** | **0.384** | **595.12** |
| 1e-4 | 6 | 0.555 | 0.273 | 640.59 |
| 3e-4 | 6 | 0.537 | 0.271 | 641.08 |
| 1e-4 | 4 | 0.509 | 0.230 | 660.28 |
| 1e-4 | 2 | 0.456 | 0.233 | 655.80 |
| 3e-4 | 2 | 0.316 | 0.166 | 738.94 |
| 3e-4 | 4 | 0.359 | 0.147 | 707.38 |
| 1e-3 | 2 | 0.233 | 0.120 | 697.80 |
| 1e-3 | 4 | 0.232 | 0.050 | 741.62 |

## C. Per-Bin Results with Full Metrics

*Table 9.* GraphSAGE test metrics stratified by post-cutoff credit transaction count bin ($N = 474$ of 10,000 test customers; the remaining 9,526 have no extra credit transactions after the cutoff and are excluded here). Per-bin $R^2$ values are lower than the overall because these bins capture the customers whose behaviour changed after the cutoff. Overall test $R^2$: Raw $= 0.692$, Agg-Simple $= 0.752$.

| Txn diff | N | $MAE_{raw}$ | $MAE_{agg}$ | $R^2_{raw}$ | $R^2_{agg}$ | $\Delta R^2$ |
|---|---|---|---|---|---|---|
| 1–5 | 280 | 533 | 576 | 0.636 | 0.529 | $-0.107$ |
| 5–16 | 76 | 560 | 602 | 0.500 | 0.340 | $-0.160$ |
| 16–688 | 44 | 711 | 319 | 0.165 | 0.608 | $+0.443$ |
| 688–15,336 | 74 | 762 | 373 | 0.063 | 0.506 | $+0.443$ |

## D. Dataset Statistics

Statistics computed on the 10,000-customer test split (temporal cutoff 2024-09-30).

*Table 10.* Transaction node-count statistics per customer. Graph = nodes visible at test cutoff; Raw = full parquet file.

| Statistic | Graph | Raw |
|---|---|---|
| Mean | 3,097 | 3,177 |
| Median | 195 | 198 |
| Std | 11,690 | 11,976 |
| p95 | 8,682 | 8,778 |
| Max | 66,991 | 88,717 |

*Table 11.* Payment node-count statistics per customer. 69% of customers have zero payment nodes (no loan/mortgage products).

| Statistic | Graph | Raw |
|---|---|---|
| Mean | 99 | 102 |
| Median | 0 | 0 |
| Std | 170 | 175 |
| p95 | 495 | 512 |
| Max | 599 | 599 |

