# OpenReview forum: "Context Window Failures in Relational Foundation Models"
_ICML.cc/2026/Workshop/FMSD — FMSD @ ICML 2026 Poster_

### Official Review · Reviewer_Mn2k · 2026-05-18

**Rating:** 5
**Confidence:** 4

**Review:**

This paper introduces a synthetic dataset designed to evaluate the neighborhood-size limitations of recent relational FM. Experiments show that RT, Griffin, and RelGT perform poorly on raw high-cardinality data (R² ≤ 0.18), while simple temporal aggregation boosts performance to R² = 0.65. The results highlight scalability issues in current relational architectures, particularly for high-frequency entities, and provide insights into challenges faced in real-world relational settings.

Strengths: Clear evaluation, practical insights, and a valuable benchmark dataset.
Weaknesses: No new method or mitigation strategy is proposed.

---

### Official Review · Reviewer_MKKS · 2026-05-22
**A Clear but Synthetic Study of Context-Window Failures**

**Rating:** 6
**Confidence:** 4

**Review:**

## Summary

This paper introduces Animus, a synthetic financial relational dataset designed to test whether graph-based relational foundation models can handle high-cardinality neighborhoods. The paper finds that several recent relational models perform poorly on the raw event-level representation, while a simple monthly pre-aggregation greatly improves performance. Overall, the paper shows an intuitive failure pattern of current relational models and presents pre-aggregation as a potential practical fix.

## Strengths

The paper studies an important and concrete failure mode. Many real relational databases contain entities with large amounts of related records, so context-window and neighborhood-budget limitations are highly relevant. The paper uses a carefully designed synthetic benchmark to verify this issue in a controlled setting.

The proposed pre-aggregation solution is also clean and intuitive. By modifying the RDB representation through monthly aggregation, the method reduces neighborhood size and makes the relevant signal easier for the model to access. Although this is a fixed solution for this benchmark, similar aggregation ideas may be useful in broader relational prediction settings.

## Areas for Improvement

The main limitation is that the benchmark is synthetic and somewhat adversarial. The task is explicitly designed so that the target is recoverable by summing many transaction values. This makes it a useful stress test, but it may overstate how often the same failure occurs in real relational prediction tasks.

For the proposed solution, the pre-aggregation strategy is not fully compared with other alternatives. For example, it would be useful to compare against simpler aggregate statistics or different aggregation designs. The paper could also analyze how the choice of aggregation granularity affects performance.

## Detailed Comments

The authors should discuss more clearly how representative Animus is of real-world high-cardinality relational prediction problems. A real-world or semi-synthetic validation would make the result more convincing.

It would be helpful to compare the monthly pre-aggregation against other feature-generation or aggregation baselines, such as global transaction statistics, category-level summaries, or different temporal windows.

I wonder whether alternative neighborhood sampling strategies could also address the issue. For example, given the long-tailed distribution of transaction records, it may be possible to design a reweighted sampling method that keeps the neighborhood size limited while better preserving the overall distribution and increasing the probability of sampling rare but informative tail events. This could help determine whether the failure is mainly due to the bounded neighborhood size itself, or due to the sampling strategy failing to capture important low-probability transaction patterns.

## Justification of Score

I would rate this paper as a 6. It proposes an intuitive and meaningful failure case, and the pre-aggregation solution is simple and practically relevant. However, my main concern is that the benchmark is synthetic and deliberately constructed around this aggregation problem, which makes the empirical result less convincing as evidence for broader real-world relational foundation model failures. The paper is relevant and useful for the workshop, but I would like stronger validation or more comparison of aggregation strategies for a higher score.